

# Prediction of pKa values using the PM6 semiempirical method

Jimmy C. Kromann, Frej Larsen, Hadeel Moustafa and Jan H. Jensen

Department of Chemistry, University of Copenhagen, Copenhagen, Denmark

## ABSTRACT

The PM6 semiempirical method and the dispersion and hydrogen bond-corrected PM6-D3H+ method are used together with the SMD and COSMO continuum solvation models to predict pKa values of pyridines, alcohols, phenols, benzoic acids, carboxylic acids, and phenols using isodesmic reactions and compared to published ab initio results. The pKa values of pyridines, alcohols, phenols, and benzoic acids considered in this study can generally be predicted with PM6 and ab initio methods to within the same overall accuracy, with average mean absolute differences (MADs) of 0.6–0.7 pH units. For carboxylic acids, the accuracy (0.7–1.0 pH units) is also comparable to ab initio results if a single outlier is removed. For primary, secondary, and tertiary amines the accuracy is, respectively, similar (0.5–0.6), slightly worse (0.5–1.0), and worse (1.0–2.5), provided that di- and tri-ethylamine are used as reference molecules for secondary and tertiary amines. When applied to a drug-like molecule where an empirical pKa predictor exhibits a large (4.9 pH unit) error, we find that the errors for PM6-based predictions are roughly the same in magnitude but opposite in sign. As a result, most of the PM6-based methods predict the correct protonation state at physiological pH, while the empirical predictor does not. The computational cost is around 2–5 min per conformer per core processor, making PM6-based pKa prediction computationally efficient enough to be used for high-throughput screening using on the order of 100 core processors.

## INTRODUCTION

A large proportion of organic molecules relevant to medicine and biotechnology contain one or more ionizable groups, which means that fundamental physical and chemical properties, such as the charge of the molecule, depend on the pH of the solution via the corresponding pKa values of the molecules. As drug- and material design increasingly is being done through high throughput screens, fast—yet accurate—computational pKa prediction methods are becoming crucial to the design process.

There are several empirical pKa prediction tools, such as ACD pKa DB (ACDLabs, Toronto, Canada), Chemaxon (Chemaxon, Budapest, Hungary), and Epik (Schördinger, New York, USA), that offer predictions in less than a second and can be used by non-experts. These methods are generally quite accurate but can fail for classes of molecules that are not found in the underlying databases. *Settimo, Bellman & Knegtel (2013)* have recently shown that the empirical methods are particularly prone to failure for amines,

Corresponding author
Jan H. Jensen, jhjensen@chem.ku.dk

which represent a large fraction of drugs currently on the market or in development. The underlying databases are not public and it is therefore difficult to anticipate when empirical methods will fail. Furthermore, the user is generally not able to augment the databases for cases where the empirical methods are found to fail.

pKa values can be predicted with significantly less empiricism using electronic structure theory (QM) (for a review see *Ho (2014)*). The accuracy of these QM-based predictions appear to rival that of the empirical approaches, but a direct comparison to empirical methods on a common set of molecules has not appeared in the literature and most QM-based pKa prediction studies have focused on relatively small sets of simple benchmark molecules. Two notable exceptions are the studies by *Klicić et al. (2002)* and *Eckert & Klamt (2005)* who computed pKa values for sets of drug-like molecules. *Klicić et al. (2002)* computed the standard free energy change for

$$BH^+ \rightleftharpoons B + H^+ \qquad (1)$$

using B3LYP/cc-pVTZ//B3LYP/6-31G(d), with diffuse functions added to negative functional groups, and the Poisson-Boltzmann continuum solvation model implemented in the Jaguar software package. The gas phase deprotonation standard free energy is computed without vibrational corrections. The pKa values are computed by

$$\text{pKa} = A\frac{\Delta G^\circ}{RT\ln(10)} + B \qquad (2)$$

where $A$ and $B$ are found by a linear fit to experimental pKa values for a training set of 200 molecules. Atomic radii for the ions used in the calculation of solvation free energies were optimized as part of the fitting procedure. When applied to the prediction of pKa values for 16 drug-like molecules, the mean absolute difference relative to experiment was 0.6 pH units.

*Eckert & Klamt (2005)* computed the standard free energy change for

$$BH^+ + H_2O \rightleftharpoons B + H_3O^+ \qquad (3)$$

using BP/TZVP and the COSMOtherm continuum solvation model. The gas phase deprotonation standard free energy is computed without vibrational corrections and the pKa values are computed using Eq. (2), where $A$ and $B$ are found by a linear fit to experimental pKa values for a training set of 43 amines. *Eckert & Klamt (2005)* observed that the method systematically underestimates the pKa of secondary and tertiary aliphatic amines by ca 1 and 2 pH units, respectively, so an additional empirical correction is added for these two molecule types. Using this approach the pKa values of 58 drug-like molecules containing one or more ionizable N atoms can be reproduced with a root mean square deviation (RMSD) of 0.7 pH units.

While quite accurate, both methods rely on DFT calculations which are computationally too expensive for routine use in high-throughput screening and design. Semiempirical QM (SQM) methods are many orders of magnitude faster than conventional QM but their application to small molecule pKa prediction has been very

limited and have focused mainly indirect prediction using atomic charges and other molecular descriptors (*Stewart, 2008*; *Rayne, Forest & Friesen, 2009*; *Ugur et al., 2014*; *Juranić, 2014*) rather than a direct prediction using relative free energies used in this study. The most likely reason for this is that semiempirical methods give significantly worse pKa predictions if used with an arbitrary reference molecule such as $H_2O$. However, several researchers (*Li, Ruiz-López & Maigret, 1997*, *Li, Robertson & Jensen, 2004*; *Govender & Cukrowski, 2010*; *Sastre et al., 2012*; *Toth et al., 2001*; *Ho & Coote, 2009*; *Ho et al., 2010*) have shown that a judicious choice of reference molecule is a very effective way of reducing the error in pKa predictions. Here we show that this approach is the key to predict accurate pKa values using PM6 and continuum solvation methods.

## COMPUTATIONAL METHODOLOGY

The pKa values are computed by

$$\mathrm{pKa} = \mathrm{pKa}^{\mathrm{ref}} + \frac{\Delta G^\circ}{RT\ln(10)} \tag{4}$$

where $\Delta G^\circ$ denotes the change in standard free energy for the isodesmic reaction

$$\mathrm{BH}^+ + \mathrm{B_{ref}} \rightleftharpoons \mathrm{B} + \mathrm{B_{ref}H}^+ \tag{5}$$

where the standard free energy of molecule X is computed as the sum of the PM6 heat of formation, the rigid rotor, harmonic oscillator (RRHO) free energy correction, and the solvation free energy

$$G^\circ(X) = \Delta H_f(X) + [G^\circ_{RRHO}(X)] + \Delta G^\circ_{solv}(X) \tag{6}$$

In some calculation the $G^\circ_{RRHO}(X)$ term is neglected, which will be indicated by an $^*$. Nominally the standard state for $G^\circ_{RRHO}(X)$ has been corrected to 1 M, but this effect cancels out for isodesmic reactions. All energy terms are computed using gas phase geometries. $\Delta H_f(X)$ is computed using either PM6 (*Stewart, 2007*) or PM6-D3H+ (*Kromann et al., 2014*) while $\Delta G^\circ_{solv}(X)$ is computed using either the SMD (*Marenich, Cramer & Truhlar, 2009*) or COSMO (*Klamt & Schüürmann, 1993*) solvation method. The PM6-D3H+ and SMD calculations are performed with the GAMESS program (*Schmidt et al., 1993*), the latter using the semiempirical PCM interface developed by *Steinmann et al. (2013)*, while the COSMO calculations are performed using MOPAC2012. The pKa of dimethylamine is also calculated at the M05-2X/6-311++G(d,p)/SMD$^*$ level of theory using Gaussian09 (*Frisch et al., 2014*). Geometry optimizations were performed in GAMESS using a convergence criterion of $5 \times 10^{-4}$ au, which is five times higher than default. In cases where imaginary frequencies were found this criterion was reduced to $1 \times 10^{-4}$ and, again, to $5 \times 10^{-5}$. Structures with imaginary frequencies found using the lowest convergence criterion were then ignored when computing the PM6-D3H+/SMD pKa values.

A conformational search was done for each molecule using Open Babel (*O'Boyle et al., 2011*) version 2.3.90 compiled from their GitHub repository. Conformations was

generated using genetic algorithm and RMSD diversity with the following settings for obabel;

```
obabel start.xyz -O finish.xyz --conformer --nconf 30 --score rmsd
--writeconformers
```

Open Babel does not consider $C-NH_2$ and $C-OH$ bonds to be rotatable, so several different start configuration for these sites were prepared manually. Similarly, new conformations due to nitrogen inversion for deprotonated secondary amines and protonated and deprotonated tertiary amines are generated manually where applicable. All start geometries are published on Figshare (*Jensen & Kromann, 2016*). When computing the pKa values the structures with the lowest free energies ($G°(X)$) are chosen.

For compound **1** (Fig. 3) Open Babel failed to find any conformations and Balloon (*Vainio & Johnson, 2007*) was used for the conformational search instead. The Balloon config file is published on Figshare (*Jensen & Kromann, 2016*).

## RESULTS AND DISCUSSION

### Comparison of pKa values predicted using PM6 and ab initio methods

*Sastre et al. (2012)* have computed the pKa values using isodesmic reactions and a several ab initio method for a variety of molecules containing six types of ionizable groups. Table 1 lists the molecules from *Sastre et al. (2012)* used in this study. The molecules in the first row are the reference molecules (ref) with the corresponding $pKa^{ref}$ value in parenthesis. Molecules containing chlorine have been eliminated because PM6 calculations for this elements involves *d*-integrals, which have not yet been implemented in GAMESS.

Columns 2–4 of Table 2 lists mean absolute differences (MADs) and maximum absolute differences (Max AD) relative to experiment for pKa values calculated by *Sastre et al. (2012)* using B3LYP and M05-2X/6-311++G(d,p) as well as the CBS-4B3* composite method (*Casasnovas et al., 2010*) and the SMD solvation method. The data shows that all three ab initio methods perform roughly equally well, with all three methods giving a MAD below 1 pH unit, with the exception of alcohols where the MAD ranges from 1.0–1.3 pH units. The Max ADs are lowest for amines (0.6–0.8 pH units) and highest for alcohols (2.3–2.9 pH units).

The fifth column lists the corresponding values computed using PM6-D3H+ with the SMD solvation method. For pyridines, alcohols, phenols, and benzoic acids the overall accuracy of PM6-D3H+ is comparable to the ab initio methods: the MADs are within 0.5 pH units of the ab initio values and while the Max ADs range from 0.4 (pyridines) to 2.4 (phenols). For carboxylic acids the results are dominated by a 3.5 pH unit error for trimethylacetic acid, without which the MAD is 1.0 pH units. Thus, different reference molecules should be used to predict pKa values for carboxylic acid groups bonded to secondary and tertiary carbons, using PM6 based methods. For amines the MAD and Max AD is 1.2 and 3.9 pH units, respectively. If only primary amines, which are most

**Table 1 List of molecules and experimental pKa values used for Table 2.** The first entry for each functional group is the reference used to compute the pKa values and the corresponding reference pKa value. The pKa values are taken from *Sastre et al. (2012)*.

| Pyridines | | Alcohols | | Carboxylic acids | |
|---|---|---|---|---|---|
| Pyridine | 5.2 | Ethanol | 15.9 | Acetic acid | 4.8 |
| 2-Methylpyridine | 6.0 | Methanol | 15.5 | Formic | 3.8 |
| 3-Methylpyridine | 5.7 | Propanol | 16.1 | Benzoic | 4.2 |
| 4-Methylpyridine | 6.0 | i-Propanol | 17.1 | Hexanoic | 4.8 |
| 2,3-Dimethylpyridine | 6.6 | 2-Butanol | 17.6 | Propanoic | 4.9 |
| 2,4-Dimethylpyridine | 7.0 | tert-butanol | 19.2 | Pentanoic | 4.9 |
| 3-Fluoropyridine | 3.0 | | | Trimethylacetic | 5.1 |
| 3-Cyanopyridine | 1.5 | | | | |
| **Amines** | | **Phenols** | | **Benzoic acids** | |
| Ethylamine | 10.6 | Phenol | 10.0 | Benzoic acid | 4.2 |
| Methylamine | 10.6 | p-Cyanophenol | 8.0 | p-Methylbenzoic | 4.4 |
| Propylamine | 10.5 | m-Cyanophenol | 8.6 | m-Methylbenzoic | 4.3 |
| i-Propylamine | 10.6 | m-Fluorophenol | 9.3 | p-Fluorobenzoic | 4.1 |
| Butylamine | 10.6 | p-Fluorophenol | 10.0 | | |
| 2-Butylamine | 10.6 | m-Methylphenol | 10.1 | | |
| tert-Butylamine | 10.6 | p-Methylphenol | 10.1 | | |
| Trimethylamine | 9.8 | o-Methylphenol | 10.3 | | |
| Dimethylamine | 10.6 | | | | |

similar to the reference compound, are considered the MAD and Max AD drops to 0.5 and 1.2 pH units, respectively. We investigate this point further in the next subsection.

The sixth column of Table 2 lists PM6-D3H+/SMD* pKa values computed with the $G^\circ_{RRHO}(X)$ term in Eq. (6) removed (denoted by the "*"). In all cases the change in MAD and Max AD is $\leq 0.2$ and 0.3 pH units, respectively. This small change is not surprising the use of isodesmic reactions and approach has been used in pKa prediction before (*Li, Robertson & Jensen, 2004*). Neglecting the dispersion correction (PM6/SMD*) has an even smaller effect on the pKa values, changing the MAD and Max AD by at most 0.1 pH units. It is important to note that the molecules used in this part of the study are relatively small and contain only one functional group. The effect of neglecting vibrational free energies and dispersion corrections may have a bigger effect on the pKa values computed for larger molecules with, for example, intramolecular interactions where both dispersion and vibrational effects can play an important role.

The final column of Table 2 lists PM6/COSMO* pKa values. The pKa values for alcohols, phenols, and benzoic acids are very similar to PM6/SMD with MAD and Max ADs changing by at most 0.1 pH units. In the case of pyridines and carboxylic acids Max AD changes by 0.5 and −1.0 pH units, respectively although this only changes the MAD by at most 0.2 pH units. In the case of pyridines the PM6/SMD* and PM6/COSMO* Max AD is observed for 2,3-dimethylpyridine and 2,4-dimethylpyridine, respectively, while in the case of carboxylic acids the Max AD is observed for trimethylacetic acid. In the case of amines, the accuracy of PM6/SMD* and

**Table 2 Mean absolute differences (MADs) and maximum absolute difference (Max AD) of predicted pKa values relative to experimental values for the molecules listed in Table 1.** CBS-4B3*, B3LYP, and M05-2X refer to predictions made by *Sastre et al. (2012)* using a modified CBS-4B3 composite method and the SMD solvation method, B3LYP/6-311++G(d,p)/SMD and M05-2X/6-311++G(d,p)/SMD, respectively.

|  | CBS-4B3*/SMD | B3LYP/SMD | M05-2X/SMD | PM6-D3H+/SMD | PM6-D3H+/SMD* | PM6/SMD* | PM6/COSMO* |
|---|---|---|---|---|---|---|---|
| **Amines** | | | | | | | |
| MAD | 0.2 | 0.4 | 0.3 | 1.2 | 1.2 | 1.3 | 0.7 |
| Max AD | 0.6 | 0.8 | 0.7 | 3.9 | 4.0 | 4.1 | 1.9 |
| MAD** | 0.2 | 0.4 | 0.3 | 0.5 | 0.6 | 0.6 | 0.6 |
| Max AD** | 0.6 | 0.8 | 0.7 | 1.2 | 1.4 | 1.4 | 1.4 |
| **Carboxylic acids** | | | | | | | |
| MAD | 0.7 | 0.7 | 0.6 | 1.4 | 1.3 | 1.2 | 1.0 |
| Max AD | 1.1 | 1.5 | 1.3 | 3.5 | 3.3 | 3.3 | 2.3 |
| **Pyridines** | | | | | | | |
| MAD | 0.5 | 0.6 | 0.6 | 0.2 | 0.3 | 0.3 | 0.4 |
| Max AD | 0.8 | 1.0 | 1.0 | 0.4 | 0.4 | 0.5 | 1.0 |
| **Alcohols** | | | | | | | |
| MAD | 1.3 | 1.0 | 1.3 | 0.7 | 0.8 | 0.8 | 0.8 |
| Max AD | 2.8 | 2.3 | 2.9 | 1.7 | 1.9 | 1.8 | 1.9 |
| **Phenols** | | | | | | | |
| MAD | 0.6 | 0.9 | 0.9 | 1.3 | 1.2 | 1.2 | 1.3 |
| Max AD | 1.7 | 2.2 | 2.1 | 2.4 | 2.5 | 2.4 | 2.4 |
| **Benzoic Acids** | | | | | | | |
| MAD | 0.4 | 0.5 | 0.3 | 0.3 | 0.3 | 0.3 | 0.3 |
| Max AD | 1.1 | 1.4 | 0.7 | 0.7 | 0.7 | 0.7 | 0.7 |

**Notes:**
* Indicates that the rigid rotor, harmonic oscillator free energy term is neglected.
** Indicates MAD and Max AD computed for primary amines only.

PM6/COSMO* is very similar for primary amines, but the error for di- and trimethylamine is reduced by 1.9 and 2.2 pH units, respectively, by using the COSMO solvation method implemented in MOPAC. To understand these differences, we look more closely at dimethylamine and compare the results to corresponding M05-2X/6-311++G(d,p)/SMD calculations, which is one of the methods used by *Sastre et al. (2012)*, but used here without the $G^\circ_{RRHO}(X)$ contribution to make the results directly comparable to PM6/SMD* and PM6/COSMO*. Both M05-2X/6-311++G(d,p)/SMD* and PM6/COSMO* yield pKa values for dimethylamine that are virtually identical in accuracy: 10.1 and 11.2 compared to the experimental value of 10.6 pH units. In the case of M05-2X/6-311++G(d,p)/SMD $\Delta E_{ele}$ (which replaces $\Delta H_f$ in Eq. (6)) and $\Delta\Delta G^\circ_{solv}$ the values are 11.4 and −10.7 kcal/mol, while the corresponding values for PM6/COSMO* are 3.5 and −4.2 kcal/mol. Taking M05-2X/6-311++G(d,p)/SMD* as a reference, the good performance of PM6/COSMO* is thus a result of significant error cancellation. The corresponding $\Delta\Delta G^\circ_{solv}$ computed using PM6/SMD* is −6.8 kcal/mol. While this value is closer to the M05-2X/6-311++G(d,p)/SMD* value it leads to worse error cancellation with the electronic energy contribution and therefore a less accurate pKa prediction (8.2 pH units).

**Table 3  Statistics for the predicted pKa values in Table 2 (labeled "Sastre") and the amines in Table 4 plus the primary amines in Table 2 (labeled "Amines").** Outliers were identified using the Modified Thompson $\tau$ method and removed prior to analysis. "std err" is the standard error of the estimate, $F$ is the Fischer statistic, $n$ the degrees of freedom, and $\tau$ cutoff is the cutoff used to determine outliers. The Sastre set has 36 data points and 34° of freedom including outliers, while the Amine set has 18 data points and 16° of freedom including outliers.

| | CBS-4B3*/SMD | B3LYP/SMD | M05-2X/SMD | |
|---|---|---|---|---|
| **Sastre** | | | | |
| Slope | 1.044 ± 0.033 | 0.991 ± 0.033 | 1.030 ± 0.036 | |
| Intercept | −0.19 ± 0.30 | 0.19 ± 0.30 | −0.05 ± 0.33 | |
| $r^2$/std err | 0.970 (0.7) | 0.968 (0.8) | 0.963 (0.8) | |
| $F(n)$ | 1006 (31) | 925 (31) | 817 (31) | |
| $\tau$ cutoff | 1.4 | 1.5 | 1.6 | |
| | **PM6-D3H+/SMD** | **PM6-D3H+/SMD***  | **PM6/SMD*** | **PM6/COSMO*** |
| **Sastre** | | | | |
| Slope | 0.973 ± 0.045 | 0.973 ± 0.047 | 0.975 ± 0.048 | 1.001 ± 0.041 |
| Intercept | 0.31 ± 0.44 | 0.36 ± 0.46 | 0.35 ± 0.46 | 0.17 ± 0.39 |
| $r^2$/std err | 0.936 (1.1) | 0.933 (1.2) | 0.933 (1.2) | 0.951 (1.0) |
| $F(n)$ | 466 (32) | 435 (31) | 421 (30) | 597 (31) |
| $\tau$ cutoff | 2.5 | 2.4 | 2.4 | 2.0 |
| **Amines** | | | | |
| Slope | 0.427 ± 0.065 | 0.498 ± 0.079 | 0.409 ± 0.048 | 0.607 ± 0.090 |
| Intercept | 6.06 ± 0.63 | 5.17 ± 0.80 | 6.20 ± 0.46 | 4.02 ± 0.91 |
| $r^2$/std err | 0.758 (0.6) | 0.726 (0.6) | 0.839 (0.5) | 0.753 (0.6) |
| $F(n)$ | 44 (14) | 40 (15) | 73 (14) | 46 (15) |
| $\tau$ cutoff | 3.0 | 2.4 | 3.1 | 1.8 |

**Note:**
* Indicates that the rigid rotor, harmonic oscillator free energy term is neglected.

In summary, the pKa values of the pyridines, alcohols, phenols, and benzoic acids considered in this study can generally be predicted with PM6 and ab initio methods to within the same overall accuracy, with average MADs for these four functional groups are 0.7–0.8 and 0.6–0.7 pH units, for the ab initio and PM6-based predictions. Similarly, the corresponding Max ADs ranges are 1.6–1.7 and 1.3–1.5 pH units, respectively. For carboxylic acids the PM6-based results are dominated by 2.3–3.5 pH unit errors for trimethylacetic acid, without which the MAD is 0.7–1.0 pH units and comparable to the corresponding ab initio results (0.6–0.7 pH units). Similarly, for amines the PM6-based results are dominated by a 1.9–4.1 pH unit errors for di- and trimethylamine, without which the MAD is 0.5–0.6 pH units and comparable to the corresponding ab initio results (0.2–0.3 pH units). For these simple molecules, dispersion corrections and vibrational free energy make a negligible contribution to the predicted pKa values.

Following *Seybold & Shields (2015)*, Table 3 summarizes the overall statistics for the predictions presented in Table 2 (labeled "Sastre") where outliers have been removed using the Modified Thompson $\tau$ method. As expected from our discussion above, the ab initio predictions are slightly better than the semiempirical results with $r^2$ and standard errors of 0.963–0.970 and 0.7–0.8 pH units compared to 0.933–0.951 and

1.0–1.2 pH units. PM6/COSMO is seen to perform slightly better than the other semiempirical approaches. The outliers are identified in Figs. 1A and 1B for the ab initio and semiempirical predictions. Trimethylamine and trimethyl acetic acid are outliers for all three SMD-based semiempirical predictions and the PM6/SMD* energy contributions are compared to the corresponding M05-2X/SMD* values to gain further insight. In both cases, the differences between PM6 and DFT is largest for the change in the gas phase deprotonation energy: 5.6 vs 1.5 and 3.7 vs 0.5 kcal/mol for trimethylamine and trimethylacetic acid, respectively.

## Secondary and tertiary amines

Here we investigate whether the accuracy of PM6-based predictions of amines can be improved by using different reference molecules for primary, secondary, and tertiary amines. Table 4 lists experimental and predicted pKa values for six secondary and tertiary amines shown in Fig. 2 using di- and tri-ethylamine as respective reference. The accuracy of the predicted pKa values for secondary amines is slightly worse compared to primary amines (Table 2): the MADs and Max ADs are 0.5–1.0 and 1.0–1.6 pH units, respectively, compared to 0.5–0.6 and 1.2–1.4 pH units. The lowest MAD and Max AD is observed for PM6/COSMO*. The contributions of vibrational and dispersion effects are larger than for primary amines, with respective changes of upto 0.8 and 0.9 pH units—both observed for diallylamine. This is presumably due to the fact that the secondary amines are structurally more different from the reference (diethylamine) than for the primary amines. For example, if piperidine is taken as a reference for the prediction of the pKa of morpholine and piperazine then the effects of vibrations and dispersion contribute at most 0.1 pH units. For the SMD-based predictions, the lowest MAD is observed for PM6-D3H+ without vibrational contributions.

The accuracy of the predicted pKa values for tertiary amines is significantly worse than for primary and secondary amines with MADs and Max ADs of 1.0–2.8 and 2.1–4.4 pH units, respectively. As observed for secondary amines, the lowest and next-lowest MAD is observed for PM6/COSMO and PM6-D3H+/SMD*. For these two methods, the largest error is observed for DABCO: 3.2 and 2.1 pH units for PM6-D3H plus;/SMD* and PM6/COSMO, respectively. With the exception of diisopropylmethylamine, both methods underestimate the pKa values, and using the 2 pH unit correction proposed by *Eckert & Klamt (2005)* reduces the MAD and Max AD to 0.7 and 1.2 for PM6-D3H+/SMD* for these molecules, although the Max AD increases to 3.8 pH units if diisopropylmethylamine is included. Alternatively, the accuracy can be improved by changing the reference molecule. For example, using quinuclidine as a reference, the pKa of DABCO is predicted to within 0.9 and 0.5 pH units using PM6-D3H+/SMD* and PM6/COSMO, respectively.

In summary, the large errors observed for secondary and tertiary amines in Table 2 (i.e. di- and tri-ethylamine) can be decreased by using di- and tri-ethylamine as a reference. The MAD and Max AD for secondary amines (0.5–1.0 and 1.0–1.6 pH units) are only a little larger than those observed for primary amines (0.5–0.6 and 1.2–1.4). The MAD and Max AD for tertiary amines (1.0–2.5 and 2.1–4.5 pH units) are significantly

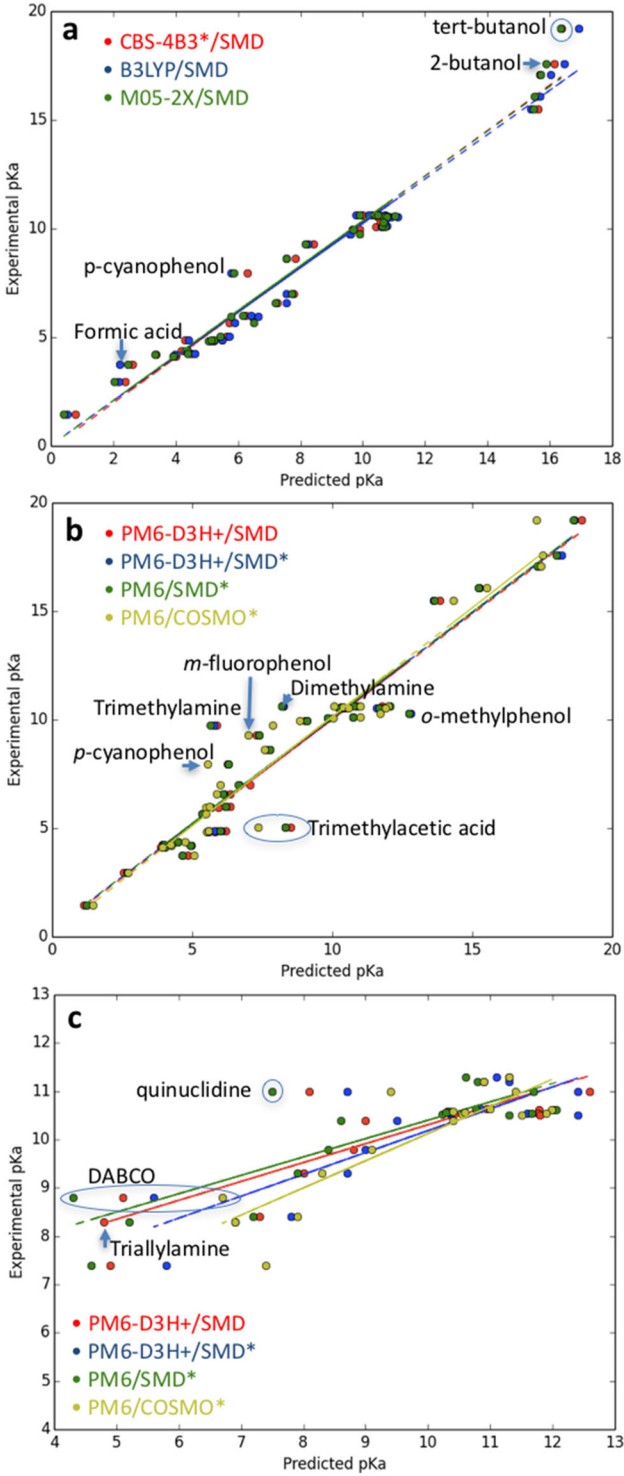

**Figure 1 Plot of (A) ab initio and (B) semiempirical pKa predictions for the molecules in Table 1 and (C) semiempirical pKa predictions for the primary amines in Table 1 and the amines in Table 4.** Outliers are identified using the Modified Thomson $\tau$ method.

**Table 4 Predicted pKa values for the secondary and tertiary amines shown in Fig. 2, using di- and tri-ethylamine as a reference, respectively.** In the case or piperazine and DABCO the pKa value corresponds to the singly protonated species.

| | Exp | PM6-D3H+/SMD | PM6-D3H+/SMD* | PM6/SMD* | PM6/COSMO* |
|---|---|---|---|---|---|
| **Secondary amines** | | | | | |
| Diethylamine | 11.1 | | | | |
| Morpholine | 8.4 | 7.3 | 7.8 | 7.2 | 7.9 |
| Piperidine | 11.2 | 10.9 | 11.3 | 10.8 | 10.9 |
| Piperazine | 9.8 | 8.8 | 9.0 | 8.4 | 9.1 |
| Pyrrolidine | 11.3 | 11.3 | 11.1 | 10.6 | 11.3 |
| Diallylamine | 9.3 | 8.0 | 8.7 | 7.9 | 8.3 |
| Diisopropylamine | 11.0 | 12.6 | 12.4 | 11.7 | 11.4 |
| MAD | | 0.9 | 0.6 | 1.0 | 0.5 |
| Max AD | | 1.6 | 1.4 | 1.4 | 1.0 |
| **Tertiary amines** | | | | | |
| Tri-ethylamine | 10.7 | | | | |
| N-methyl morpholine | 7.4 | 4.9 | 5.8 | 4.6 | 7.4 |
| Quinuclidine | 11.0 | 8.1 | 8.7 | 7.5 | 9.4 |
| DABCO | 8.8 | 5.1 | 5.6 | 4.3 | 6.7 |
| N-Ethylpyrrolidine | 10.4 | 9.0 | 9.5 | 8.6 | 10.4 |
| Triallylamine | 8.3 | 4.8 | 6.9 | 5.2 | 6.9 |
| Diisopropylmethylamine | 10.5 | 11.8 | 12.4 | 11.3 | 11.5 |
| MAD | | 2.5 | 1.9 | 2.7 | 1.0 |
| Max AD | | 3.7 | 3.2 | 4.5 | 2.1 |

**Note:**
* Indicates that the rigid rotor, harmonic oscillator free energy term is neglected.

larger than those observed for primary amines and secondary amines. As observed by *Eckert & Klamt (2005)* the pKa values tend to be underestimated and the error can be reduced somewhat by adding a 2 pH unit correction factor. Alternatively, the error can be reduced for individual molecules by choosing reference molecules with similar structures. PM6/COSMO* results in the lowest errors, followed by PM6-D3H+/SMD* for both secondary and tertiary amines.

Table 3 summarizes the overall statistics for the primary amines in Table 2 and the amines in Table 4 (labeled "Amines") where outliers have been removed using the Modified Thompson $\tau$ method. As expected from our discussion above, the predictions for amines are significantly worse than for the molecules in Table 1. In particular, the slopes deviate significantly from 1.0 and the intercept is in the range 4.0–6.2 pH units, due in part to the underestimated pKa values of tertiary amines. Interestingly, the standard error is 0.5–0.6 pH units, suggesting that reasonably accurate pKa predictions may be possible with the chosen reference molecules if the slope and intercept determined here are transferable to other systems. The outliers are identified in Fig. 1C. DABCO is an outlier for all four semiempirical predictions and the PM6/SMD* energy contributions are compared to the corresponding M05-2X/SMD* values to gain further insight. Again, the

**Figure 2** Depiction of the secondary and tertiary amines used in this study.

difference between PM6 and DFT is largest for the change in the gas phase deprotonation energy: 5.9 vs 4.2 kcal/mol.

## Application to a drug-like molecule

We explore the effect of using different reference molecules further for compound **1** shown in Fig. 3. Settimo, Bellman & Knegtel (2013) have shown that the Chemaxon pKa predictor predicts a pKa value of 9.1 for compound **1**, which is significantly higher than the experimental value of 4.2, i.e. Chemaxon predicts that **1** is charged as physiological pH when, in fact, it is neutral. Table 5 list the pKa values for **1** predicted using PM6-based methodologies using three different reference molecules (cf. Table 3). The absolute errors range from 1.7–8.5 with the error being smallest for PM6/SMD using tri-ethylamine as a reference. Given the size of compound **1** we expect that dispersion effects will make important contributions to intramolecular interactions and the difference in pKa values predicted with PM6-D3H+/SMD* and PM6/SMD* is indeed substantial (9.2–10.5 kcal/mol). The low error observed for PM6/SMD* is therefore likely fortuitous and, indeed, the error increases for reference molecules more closely related to **1**, while the opposite is seen for PM6-D3H+/SMD(*). Furthermore, the PM6-D3H+/SMD(*) results are consistent with the near systematic

**Figure 3 The structure of compound 1, heliotridane, and benzylpyrolidine.**

**Table 5 Predicted pKa values for compound 1 shown in Fig. 3, using tri-ethylamine, heliotridane, and benzylpyrrolidine as a reference, respectively.** The pKa values of heliotridane, and benzylpyrro­lidine are taken from *Morgenthaler et al. (2007)*. Note that the latter is estimated and not measured experimentally.

|  | $pKa^{ref}$ | PM6-D3H+/SMD | PM6-D3H+/SMD* | PM6/SMD* | PM6/COSMO* |
|---|---|---|---|---|---|
| Tri-ethylamine | 10.7 | −4.3 | −3.6 | 5.9 | −0.2 |
| Benzylpyrrolidene | 8.9 | −1.9 | −1.5 | 7.8 | 0.1 |
| Heliotridane | 11.4 | −1.6 | −1.8 | 8.7 | 0.7 |

**Note:**
* Indicates that the rigid rotor, harmonic oscillator free energy term is neglected.

pKa-underestimation observed for the tertiary amines in Table 4 and if the 2 pH unit correction suggested by *Eckert & Klamt (2005)* is used the error decreases to 3.7–4.1 pH units when benzylpyrrolidine or heliotridane are used as references. While these errors are substantial they lead to the correct qualitative prediction that **1** is neutral at physiological pH. However, whether PM6-based pKa predictions are sufficiently accurate to be useful in drug-design will require a great deal of additional study (see Summary and Outlook section for further information).

The computational cost of computing the free energy of a single conformation of **1** is ca 5 min on a single Intel Xeon 2.67 GHz X5550 core processor with the time roughly equally split between geometry optimization and vibrational frequency calculations. Thus, if the vibrational contributions to the standard free energy can be neglected the time requirement drops to 2–3 min per conformer per core processor. For **1** we computed the free energy of roughly 200 conformers. Thus, PM6-based pKa prediction is computationally efficient enough to be used for high throughput screening using on the order of 100 core processors.

## SUMMARY AND OUTLOOK

The PM6 semiempirical method and the dispersion and hydrogen bond-corrected PM6-D3H+ method are used together with the SMD and COSMO continuum solvation models to predict pKa values of pyridines, alcohols, phenols, benzoic acids, carboxylic acids, and phenols using isodesmic reactions. The results are compared to ab initio results published by *Sastre et al. (2012)*.

The pKa values of the pyridines, alcohols, phenols, and benzoic acids considered in this study can generally be predicted with PM6 and ab initio methods to within the same overall accuracy, with average MADs for these four functional groups of 0.7–0.8 and 0.6–0.7 pH units, for the ab initio and PM6-based predictions. Similarly, the corresponding Max ADs ranges are 1.6–1.7 and 1.3–1.5 pH units, respectively. For carboxylic acids the PM6-based results are dominated by 2.3–3.5 pH unit errors for trimethylacetic acid, without which the MAD is 0.7–1.0 pH units and comparable to the corresponding ab initio results (0.6–0.7 pH units). Similarly, for amines the PM6-based results are dominated by a 1.9–4.1 pH unit errors for di- and trimethylamine, without which the MAD is 0.5–0.6 pH units and comparable to the corresponding ab initio results (0.2–0.3 pH units). For these simple molecules, dispersion corrections and vibrational free energy make a negligible contribution to the predicted pKa values.

The large errors observed for secondary and tertiary amines in Table 2 (i.e. di- and tri-ethylamine) can be decreased by using di- and tri-ethylamine as a reference. The MAD and Max AD for secondary amines (0.5–1.0 and 1.0–1.6 pH units) are only a little larger than those observed for primary amines (0.5–0.6 and 1.2–1.4). The MAD and Max AD for tertiary amines (1.0–2.5 and 2.1–4.5 pH units) are significantly larger than those observed for primary amines and secondary amines. As observed by *Eckert & Klamt (2005)*, the pKa values tend to be underestimated and the error can be reduced somewhat by adding a 2 pH unit correction factor. Alternatively, the error can be reduced for individual molecules by choosing reference molecules with similar structures. PM6/COSMO results in the lowest errors, followed by PM6-D3H+/SMD* for both secondary and tertiary amines.

When applied to a drug-like molecule where the empirical pKa predictor from Chemaxon exhibits a large error, we find that the error is roughly the same in magnitude but opposite in sign. As a result, most of the PM6-based methods predict the correct protonation state at physiological pH, while the empirical predictor does not. The computational cost is around 2–5 min per conformer per core processor making PM6-based pKa prediction computationally efficient enough to be used for high throughput screening using on the order of 100 core processors.

While the accuracy found for PM6-based pKa prediction is encouraging, the performance needs to be tested for a much larger set of molecules with larger pKa shifts. However, several steps need to be automated to make this feasible. Many conformational search algorithms do not consider C-NH2 and C-OH single bonds rotatable and will leave the start orientation, which is often arbitrarily assigned, unchanged and this can lead to relatively large errors in the predicted pKa values. If such a conformational search algorithm is employed, one needs to prepare all possible start conformations for these sites. Similarly, conformational search algorithms do not consider inversion of secondary and tertiary amines meaning that all possible start conformations of deprotonated secondary amines and deprotonated and protonated tertiary amines must be prepared. For molecules with several ionizable sites all relevant combinations of protonation states must be generated and apparent pKa values must be extracted from the calculations.

Finally, a library of reference molecules and their experimental pKa values must be created and the most suitable reference molecules must be identified for each ionizable site in the target molecule. Work on all these steps are either currently ongoing or in the planning stages (*Jensen, 2015*).

### Funding

JCK received support from the University of Copenhagen. The funders had no role in study design, data collection and analysis, decision to publish, or preparation of the manuscript.

### Competing Interests

The authors declare that they have no competing interests.

### Author Contributions

- Jimmy C. Kromann conceived and designed the experiments, performed the experiments, analyzed the data, wrote the paper, prepared figures and/or tables, reviewed drafts of the paper.
- Frej Larsen performed the experiments, analyzed the data, reviewed drafts of the paper.
- Hadeel Moustafa performed the experiments, reviewed drafts of the paper.
- Jan H. Jensen conceived and designed the experiments, analyzed the data, wrote the paper, prepared figures and/or tables, reviewed drafts of the paper.

### Data Deposition

Figshare, https://dx.doi.org/10.6084/m9.figshare.c.3259513.v1: a list of pKa values used for Table 2, all input and output files, a config file for the Balloon program used in the conformational search, various submit and analysis scripts.

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
