# Peer review of "Prediction of pKa values using the PM6 semiempirical method"

_PeerJ, doi:10.7717/peerj.2335_

## Round 0.1 · original submission · Minor Revisions

Please adress all queries by the reviewers.

Personal comments from the editor:

Table 4 shows dramatic differences between PM6-D3H+ and PM6 although the previous tables did not show very large differences between both semiempirical methods. Please discuss this.

How do the errors in PM6 or PM6-D3H+ gas-phase protonation energies (vs. experiment or high level computation) change when moving from primary to secondary and tertiary amines? I believe that the addition of a table with these data (with each tested amine treated separately) would be very helpful for the readers and future practitioners.

Reviewer 1 ·

Basic reporting

This is an interesting manuscript, but a frustrating aspect is that the experimental pKa values used for comparison are not included for most compounds. These could easily be added to Table 1. In fact, the best solution would be to modify Table 2 to give the calculated pKa values from the various methods along with the experimental values. Also, the authors should refer somewhere to the very relevant PM6 pKa calculations by Jimmy Stewart given in http://openmopac.net/pKa_table.html.

Experimental design

In general this work is properly designed.

Validity of the findings

It would be very helpful if the authors would provide a figure comparing the calculated and experimental values, and include in the text the relevant equation with proper statistics (n, r2, s, F) along with the uncertainties for the slope & intercept. (See, e.g., the book by Shields & Seybold on this topic, or their WIRES article.)

Additional comments

After improvements, this manuscript will be of interest to many people attempting to calculate pKas, especially those dealing with high throughput applications.

Reviewer 2 ·

Basic reporting

The paper is well-written and organised in a manner that was easy to read. I did find the background / literature research on the short side. Specifically, the isodesmic or proton exchange scheme was developed quite some time ago by various groups . See for example: (a) http://dx.doi.org/10.1063/1.1337862 (b) 10.1021/ct800335v and (c) 10.1021/jp107890p. These studies have laid out quite clearly the effectiveness of an isodesmic scheme for error cancellation, as well as its limitations (e.g. the need for a structurally similar reference with accurately known pKas). Another minor point is there should be a footnote to explain what "**" in Table 2 means.

Experimental design

The research question is well-defined, namely whether contemporary semi-empirical methods can provide cost-effective predictions of pKas. I do have a number of suggestions for improvement:

(1) Computational methods: It was not clear how the solvation free energies were computed - e.g. were these done on gas phase or solution phase optimised geometries? Strictly speaking, the gas and solution phase components of the solvation free energy should be computed on geometries optimised in the respective phases. How sensitive are the results to this choice?

(2) There is a lot of data condensed into the Tables which could actually be used to provide even deeper insights. For example, I would love to see a breakdown of the solution phase energies into the gas phase and solvation contributions as laid out in eqn (6). This would be useful for identifying the sources of errors especially for the outliers.

(3) The dataset molecules in Table 1 are structurally very similar (the substituents are mostly aliphatic groups). It would be interesting to see a more diverse selection of molecules (e.g. EWG and EDG) as the authors alluded to in their conclusion.

Validity of the findings

I think the conclusions are fair based on the results presented. However, I do recommend the authors consider my earlier suggestions to provide clearer insights as to why semi-empirical methods can sometimes fail badly even for isodesmic reactions. This will spur further research into improving these methods.

Reviewer 3 ·

Basic reporting

- Line 132: change "can play and important role" to "can play an important role"
- Citations need to conform to the journal style thoughout: see, e.g., "taken from (Morgenthaler et al., 2007)" in Table 4 caption should be changed to "taken from Morgenthaler et al. (2007)"
- References in the bibliography need to be consistently formatted to journal guidelines
- Other groups have reported validation efforts for predicting pKa values using the PM6 method (see, e.g., Rayne et al. [2009], Juranić [2014], etc.). The authors should cite and incorporate the findings of all these prior PM6 pKa validation efforts into the current study to demonstrate an understanding of the prior literature in this field. Otherwise, it looks as though the authors are attempting to make their research appear more novel than it actually is.

Experimental design

- Experimental design is appropriate.

Validity of the findings

- The findings appear valid.

---

## Round 0.2 · Minor Revisions

Thank you for your efforts at addressing the reviewer's comments. In spite of that, the reviewers (and myself) still think that additional data should be moved from the Supporting Information to the main text as tables/graphs. Specifically:

-per reviewer 1's request, please include the ref. pKa data in table 1. The Supporting Material deposited in figshare is very complete, but it will enormously help the reader (and make your paper much more persusive at first reading) if the most salient pieces were included in the paper itself.

- contra reviewer 1's comment, I do acknowledge that p.799 of the quoted Stewart (2008) reference describes the pKa computation procedure which generated the data present in http://openmopac.net/pKa_table.html. This method (also used by Rayne) as well as the method by Juranic, however, do not computes pKa from the energy difference itself, but from an empirical fit of the O-H bond distances and approximate charges (or N and H charges, plus a dummy variable stating whether the amine is primary, secondary or terciary, for Juranic, 2014). These pKa computation approaches are therefore fundamentally different from the one used in your paper. Your references to this literature in the introduction, however, do not make this clear enough. Please improve this to clearly compare the competing methods for PM6-based pka computations to the your approach.

-Do include the statistical data regarding slope, R-squared and outliers. A motivated reader may easily graph the data you have computed (and which are present in the spreadsheet referred to in your figshare area), but your explanation and discussion would be much more readable, and certainly more persuasive, if you included those graphs, slopes and correlation coefficientes in the paper. That analysis shows more clearly than the aggregat tables exactly where PM6 affords better correlation/slope that even CBS-4B3 (pyridines), the identity of the outliers, how poorly all methods (even CBS-4B3) correlate to experimental pKa in amines (in spite of a seemingly low 0.2 MAD for CBS-4B3), etc.

-per reviewer 2's request (and also related to my previous request which I may not have worded clearly) please add data regarding the likely origin of the errors in the outliers: do they come from gas phase energies or the solvation? A simple comparison of the B3LYP gas-phase energy changes (on PM6-optimized geometries, to reduce computational effort) and solvation effects might be enough to tell whether the gas-phase acidities (and/or solvation) of PM6 generally track the DFT results.

Reviewer 1 ·

Basic reporting

.

Experimental design

.

Validity of the findings

.

Additional comments

Reviewer Comments – Reviewer 1
The authors have not adequately responded to any of the concerns raised in my original review. My original comments are shown first. The authors’ responses are shown next; and my further responses to them are shown below.
(1) Basic reporting
This is an interesting manuscript, but a frustrating aspect is that the experimental pKa values used for comparison are not included for most compounds. These could easily be added to Table 1. In fact, the best solution would be to modify Table 2 to give the calculated pKa values from the various methods along with the experimental values.

Authors: The values are already provided in Supplementary Materials

The copy I received contained no reference at all to “Supplementary Materials”. If these materials are available directions for accessing them should be clearly presented in the normal position just before the References.

(2) Also, the authors should refer somewhere to the very relevant PM6 pKa calculations by Jimmy Stewart given in http://openmopac.net/pKa_table.html.

Authors: We already refer to this approach in the introduction (Stewart 2008).

The reference Stewart (2008) concerns proteins and has nothing at all to do with pKa estimates. As clearly indicated, the relevant Stewart study is not a formal publication, but has been made widely available to workers by Stewart on the web page as indicated. Apparently the authors didn’t even bother to look at it.

(3) Validity of the findings
It would be very helpful if the authors would provide a figure comparing the calculated and experimental values, and include in the text the relevant equation with proper statistics (n, r2, s, F) along with the uncertainties for the slope & intercept. (See, e.g., the book by Shields & Seybold on this topic, or their WIRES article.)

Authors: The statistical analysis the reviewer refers to is done in the context of a QSAR prediction of pKa from QM data, i.e. to gauge the accuracy a linear fit to be used in the prediction of unknown pKa values. The statistics used in this paper is just aimed at gauging the accuracy of the predicted values and, in our opinion, is more than adequate for the task. If the reviewer can explain how the requested statistics is to be used in the context of the current paper we will be happy to reconsider the request.

This is a standard way to compare not just QSAR results, but any studies in this field. It would be helpful, and I don’t understand the authors’ reluctance to include it.

Annotated reviews are not available for download in order to protect the identity of reviewers who chose to remain anonymous.

Reviewer 2 ·

Basic reporting

As before.

Experimental design

As before.

Validity of the findings

As before.

Additional comments

I thank the authors for the revised manuscript. I would still like the authors to address my second point as to what is the major source of error in these calculations, especially for the outliers. Is it the gas phase energies, or the solvation component?

---

## Round 0.3 · accepted · Accept

I am pleased with the changes. I believe there is a mistake in line 170:

"As expected from our discussion above, the ab initio predictions are slightly worse than the semiempirical results with r2 and standard errors of 0.963 - 0.970 and 0.7 - 0.8 pH units compared to 0.933
172 - 0.951 and 1.0 - 1.2 pH units"

Shouldn't that be "As expected from our discussion above, the ab initio predictions are slightly BETTER than the semiempirical results with r2 and standard errors of 0.963 - 0.970 and 0.7 - 0.8 pH units compared to 0.933 172 - 0.951 and 1.0 - 1.2 pH units"? Please arrange for this correction with PeerJ's technical staff